# Inflammation as a Sex-Specific Mediator in the Relationship between Maternal and Offspring Obesity in C57Bl/6J Mice

**DOI:** 10.3390/biology13060399

**Published:** 2024-05-31

**Authors:** Lauren A. Buckley, Debra R. Kulhanek, Adrienne Bruder, Tate Gisslen, Megan E. Paulsen

**Affiliations:** 1Department of Pediatrics, Division of Neonatology, University of Minnesota Medical School, Minneapolis, MN 55454, USA; kulha012@umn.edu (D.R.K.); brudera@med.umich.edu (A.B.); tgisslen@umn.edu (T.G.); megan.paulsen@childrensmn.org (M.E.P.); 2Masonic Institute for the Developing Brain, University of Minnesota, Minneapolis, MN 55414, USA

**Keywords:** inflammation, obesity, fetal development, hypothalamus, energy metabolism, sex differences

## Abstract

**Simple Summary:**

The developmental origins of health and disease (DOHaD) hypothesis posits that stressful exposures during early development alter biological processes and increase the risk for chronic disease later in life. A widely accepted example of the DOHaD hypothesis is the strong link between exposure to maternal obesity and the development of obesity in offspring. Mechanisms explaining this relationship are not fully understood. Therefore, this study investigates both the hypothalamus and inflammation as important variables in the relationship between maternal and offspring obesity. The hypothalamus is the master regulator of energy homeostasis in the body. Inflammation in the hypothalamus is associated with the development of obesity. We used an established mouse model of maternal obesity to study body composition, energy homeostasis, and inflammation in male and female offspring. Like in our previous work, both male and female offspring exposed to maternal obesity had a phenotype consistent with energy excess. We found decreased markers of inflammation in the body and hypothalamus of offspring exposed to maternal obesity compared with offspring exposed to a control pregnancy. Inflammation was different between male and female offspring. Future studies focused on understanding how inflammation affects hypothalamic development in males and females may provide neuroprotective strategies to improve hypothalamic function thereby decreasing obesity risk.

**Abstract:**

Maternal obesity is a well-established risk factor for offspring obesity development. The relationship between maternal and offspring obesity is mediated in part by developmental programming of offspring metabolic circuitry, including hypothalamic signaling. Dysregulated hypothalamic inflammation has also been linked to development of obesity. We utilized an established C57Bl/6J mouse model of high-fat, high-sugar diet induced maternal obesity to evaluate the effect of maternal obesity on systemic and hypothalamic TNF-α, IL-6, and IL-1β levels in neonatal and adult offspring. The offspring of dams with obesity demonstrated increased adiposity and decreased activity compared to control offspring. Maternal obesity was associated with decreased plasma TNF-α, IL-6 and IL-1β in adult female offspring and decreased plasma IL-6 in neonatal male offspring. Neonatal female offspring of obese dams had decreased TNF-α gene expression in the hypothalamus compared to control females, while neonatal and adult male offspring of obese dams had decreased IL-6 gene expression in the hypothalamus compared to control males. In summary, our results highlight important sex differences in the inflammatory phenotype of offspring exposed to maternal obesity. Sex-specific immunomodulatory mechanisms should be considered in future efforts to develop therapeutic interventions for obesity prevention and treatment.

## 1. Introduction

The obesity epidemic is a major public health crisis affecting the reproductive health of millions of women [1]. In the United States, nearly 40% of women of reproductive age are now classified as obese [2]. In addition to higher rates of pregnancy complications, offspring born to mothers with obesity have increased risk of obesity and metabolic syndrome later in life, perpetuating a transgenerational cycle [1,2,3,4]. Specifically, children born to obese women have increased body weight and adiposity at birth and higher BMI and blood pressure later in childhood compared to offspring of non-obese women [1,3,4,5]. These findings persist even when correcting for postnatal diet and lifestyle factors [4,5]. 

Animal models have repeatedly shown that maternal obesity causes increased adiposity and alters adipose tissue function in offspring [3]. Within the hypothalamus, maternal obesity impacts the circuitry responsible for the regulation of hunger, satiety, and energy expenditure [3,6]. While it is clear that maternal obesity affects the developmental programming of metabolic circuitry in offspring, successful therapeutic interventions to disrupt the transmission of obesity from a mother to her offspring are yet to be identified [1,3,6,7,8,9,10].

The neonatal–perinatal medicine field has applied neuroprotective strategies to infants born premature and term infants who have sustained developmental injury such as hypoxic–ischemic injury [11,12]. While these interventions have not yet been studied in offspring born to mothers exposed to maternal obesity, our group is interested in the potential of this application. The hypothalamus, which plays a key role in energy homeostasis through the regulation of energy intake and expenditure, is highly sensitive to injury during key periods of development [13,14,15]. Likewise, models of hypothalamic injury cause obesity. We have previously shown that a maternal high-fat, high-sugar diet increases adiposity, decreases energy expenditure, and decreases neuronal activity and oxidative energy production in the hypothalamus of adult mouse offspring, suggesting developmental injury [6]. Specific mechanisms that mediate the relationship between exposure to maternal obesity and abnormal hypothalamic function in offspring remain unclear.

We hypothesize that a potential mediating variable between maternal and offspring obesity is inflammatory-mediated hypothalamic injury. Obesity is an inflammatory state, and dysregulated inflammation within the hypothalamus has been implicated in the development of obesity in adults [16,17,18,19,20]. The role of hypothalamic inflammation in the transmission of obesity from a mother to her offspring has received limited study [21,22,23,24,25]. However, previous work by our group established that early inflammatory stress contributes to long-term transcriptomic and behavior changes in rodent hippocampus [26,27,28], highlighting that early inflammatory-mediated injury can cause long-term changes within the developing brain. 

The aim of this study was to investigate the relationships between maternal obesity, offspring metabolism, and hypothalamic inflammation. We utilized an established mouse model of diet-induced maternal obesity [29] to characterize offspring metabolic phenotypes, as well as systemic and hypothalamic inflammation. TNF-α, IL-1β, and IL-6 were chosen as inflammatory targets as they are most commonly associated with and reported in the obesity literature [16,25,30,31,32,33,34,35,36,37].

## 2. Materials and Methods

### 2.1. Animal Model

C57Bl/6J mice (The Jackson Laboratory, Bar Harbor, ME, USA) were group-housed under standard conditions in an animal care facility with 12 h light: dark cycles. At approximately 6 weeks of age, females were randomized to a control diet (CON; D12489B, Research Diets, New Brunswick, NJ, USA, 10.6 kcal% fat, 16.8 kcal% protein, 72.6 kcal% carbohydrate, 240 g/kg sucrose) or high-fat-high-sucrose obesogenic (OB) diet. The OB diet included a high-fat pellet (Western Diet D12079B, Research Diets, New Brunswick, NJ, USA, 40.0 kcal% fat, 17.0 kcal% protein, 43.0 kcal% carbohydrate, 340 g/kg sucrose) and 20% sucrose solution supplemented with vitamins (AIN Vitamin Mixture; MP Biomedicals, Solon, OH, USA) and minerals (AIN-93M Mineral Mix; MP Biomedicals, Solon, OH, USA) [6,9,29]. All animals were fed ad libitum and had free access to water. When OB females gained 25% of their body weight (approximately 10–12 weeks of age), they were age-matched with CON females and mated with males maintained on a standard chow (Teklad Global 18% Protein Rodent Diet 2918, ENVIGO, Madison, WI, USA; 18 kcal% fat, 24 kcal% protein, 58 kcal% carbohydrate). Dams were maintained on CON or OB diet throughout pregnancy and lactation because hypothalamic development in mice occurs during both the embryonic and early postnatal periods [38]. Dam macronutrient intake from pregnancy through lactation was estimated by weighing food pellet and sucrose solutions every 48 h.

Offspring were delivered spontaneously and studied during the neonatal period on postnatal day 9 (PN9) and in adulthood at approximately 4 months of age. Litters were randomly culled to equal size at PN9 (4 pups/L). The offspring utilized in this study were a subset from a larger study, and thus all offspring underwent intraperitoneal injection of normal saline (20 μL) on PN7. Offspring were weaned on PN21 and maintained on standard chow.

### 2.2. Blood and Tissue Collection

Blood and tissue collection from fasted animals were performed as previously described [6,9]. Dams and offspring were euthanized by rapid decapitation. Whole blood was collected by truncal blood sampling (offspring) and cardiac puncture (dams). Whole blood glucose was measured via glucometer (Contour Next, Ascensia Diabetes Care, Parsippany, NJ, USA). Plasma was isolated by spinning whole blood at 5000 RPM for 10 min; supernatant was decanted and stored at −80 °C. Offspring whole brains were removed, and the hypothalamus was rapidly dissected on ice, flash-frozen in liquid nitrogen, and stored at −80 °C until the time of analysis.

### 2.3. Plasma Analyses

Plasma cytokines (TNF-α, IL-1β, and IL-6) were quantified by the Cytokine Reference Laboratory at the University of Minnesota using automated multiplex ELISA per manufacturer’s instructions (Ella, Simple Plex, ProteinSimple, San Jose, CA, USA). Plasma cholesterol, triglycerides, phospholipids, non-esterified fatty acids (NEFA), insulin, and leptin were measured at the Mouse Metabolic Phenotyping Center at the University of Cincinnati by standard protocol as previously described [9].

### 2.4. Hypothalamic Gene Expression Analyses

RNA was isolated from whole hypothalamus (Rneasy Mini Kit, Qiagen, Redwood City, CA, USA) and converted to cDNA (High-Capacity RNA-to-cDNA Kit, Applied Biosystems, Waltham, MA, USA). Relative gene expression was measured using Real-Time qPCR with TaqMan primers (Appendix A, ThermoFisher Scientific, Carlsbad, CA, USA). Gene expression was normalized to 18S rRNA using the cycle threshold (ΔΔCT) method. We have previously confirmed 18S rRNA (M-value < 1.5) as a suitable reference gene between groups [8,39].

### 2.5. In Vivo Physiology Analyses

Fat mass, fat free mass, and adiposity (fat mass/body weight, %) were measured using Echo-MRI (EchoMRI, Echo Medical Systems, Houston, TX, USA) as previously described [6,9]. Energy intake was calculated based on the amount of chow consumed, as measured by BioDAQ episodic Food Intake Monitor for mice (Research Diet, New Brunswick, NJ, USA). Basal metabolic rate (BMR) was measured using indirect calorimetry in singly housed freely moving mice after a period of acclimation (Oxymax, Columbus Instruments, Columbus, OH, USA) and calculated as previously described by our group and others [6,9,40,41]. Activity levels were measured using automated singly housed cages after a period of acclimation (Oxymax, Columbus Instruments, OH, USA).

### 2.6. Statistical Analysis and Data Presentation

The exposure variable for this study was maternal diet (CON, OB), and experimental numbers (N/group) were determined using our group’s previous work [6,8,9]. Group differences were measured by unpaired *t*-test. As there are recognized differences in metabolic outcomes and response to inflammation in males and females, we analyzed the sexes separately. *p*-values < 0.05 were considered statistically significant. Data are represented as mean ± SEM, with N representing the number of animals per group. Statistical analysis and graphics were performed using GraphPad Prism version 9 (GraphPad Software, Inc., La Jolla, CA, USA).

## 3. Results

### 3.1. Dam Characteristics

The characteristics of dams fed CON and OB diets are shown in Figure 1. OB dams weighed 8% more than CON dams at mating (*p* = 0.046) and weighed 10% more than CON dams at weaning (Figure 1a, *p* = 0.002). To determine the effect of maternal diet on maternal systemic inflammation, key plasma cytokines implicated in obesity (TNF-α, IL-6, IL-1β) were measured in dams at weaning (PN21). There were no differences in plasma cytokine levels between CON and OB dams (Figure 1b–d).

### 3.2. Offspring Phenotype

The characteristics of the offspring energy balance are shown in Figure 2. Adult male offspring in the OB group were 7% heavier than in CON males (Figure 2a, *p* = 0.04). Body weights did not differ between OB and CON females. Adiposity was 28% higher in OB females versus CON females (Figure 2b, *p* = 0.04), and there was a non-significant trend towards increased adiposity in OB males compared to CON males (Figure 2b, *p* = 0.06). Energy intake and BMR were not affected by maternal diet or offspring sex (Figure 2c,d). Offspring activity level was 25% lower in OB females compared to CON females (Figure 2e, *p* = 0.04). Activity level did not differ between OB and CON males.

The results of offspring metabolic profile are shown in Table 1. Insulin, leptin, NEFA, cholesterol, triglyceride, and phospholipid levels were measured in plasma at 4 months of age. The plasma metabolic profile did not differ between CON and OB offspring.

### 3.3. Offspring Inflammation

To determine the impact of maternal diet on offspring systemic inflammation, plasma cytokine levels were measured in neonatal and adult offspring (Figure 3). Plasma IL-1β was decreased in OB adult female offspring compared to CON adult female offspring (Figure 3a, *p* = 0.03). There was a nonsignificant trend towards decreased plasma TNF-α in OB adult female offspring compared to CON adult female offspring (Figure 3b, *p* = 0.08). There were nonsignificant trends toward decreased IL-6 in OB neonatal male offspring and OB adult female offspring compared to their respective controls (Figure 3c, males *p* = 0.05; females *p* = 0.08). Plasma cytokines in female neonatal offspring and adult male offspring were not affected by maternal diet (Figure 3a–c).

To determine whether maternal OB diet causes inflammatory changes within the hypothalamus, gene expression of TNF-α, IL-6, and IL-1β were measured in the hypothalamic tissue of neonatal and adult offspring (Figure 4). TNF-α expression was lower in the female OB offspring compared to the CON offspring hypothalamus (Figure 4a, p = 0.04), and there was a non-significant trend towards lower TNF-α in neonatal male offspring hypothalamus (a, *p* = 0.08). IL-6 expression was lower in neonatal and adult OB male offspring hypothalamus compared to CON offspring hypothalamus (b, *p* = 0.02; *p* = 0.01). There was a nonsignificant trend towards lower IL-1β expression in the adult OB offspring hypothalamus compared to the CON offspring hypothalamus (c, *p* = 0.07).

## 4. Discussion

The objective of this study was to better understand the role of inflammation in the developmental origins of obesity. The main finding of the study is that male and female offspring exposed to maternal obesity have different inflammatory phenotypes despite a common metabolic phenotype consistent with energy excess. We found that offspring exposed to maternal diet-induced obesity have decreased plasma levels and downregulated hypothalamic gene expression of TNF-α, IL-6, and IL-1β. In the following paragraphs, we examine our results in the context of the existing literature, review the study’s limitations, and discuss key insights relevant to the development of targeted obesity prevention strategies.

Contrary to our hypothesis, we did not observe elevated plasma cytokines in male or female offspring of obese dams. We suspect that systemic inflammation was not increased in our study due to the specific metabolic phenotype observed in offspring exposed to maternal obesity. The excess secretion of TNF-α, IL-6, and IL-1β from adipocytes may cause chronic and low-grade inflammation involved in obesity-associated systemic and hepatic insulin resistance [33,42,43,44]. However, our mouse model does not cause significant systemic insulin resistance in OB offspring [6,9]. Offspring exposed to maternal obesity in our study had normal leptin levels in contrast to previous work demonstrating elevated leptin and insulin levels in offspring exposed to maternal obesity [21,22]. These findings suggest a comparatively milder metabolic phenotype produced by our mouse model. In addition, our cohort was fed a standard diet (vs. a obesogenic diet associated with inflammation) and was studied in the fasting state, which may have contributed to the non-elevation of inflammatory markers observed in offspring exposed to maternal obesity despite having higher fat mass. Importantly, we observed phenotypic and hypothalamic inflammatory changes in adult offspring despite an absence of systemic inflammation or altered insulin and leptin levels. These findings indicate that lasting hypothalamic injury occurs even in the absence of ongoing exposure to dietary excess or overt metabolic derangements.

Consistent with previous work by our group, we found lower activity levels in female offspring exposed to maternal obesity [6,9]. We postulate that decreased activity contributes to decreased plasma TNF-α, IL-6, and IL-1β levels observed in these animals. Exercise is a known inducer of TNF-α, IL-6, and IL-1β [45,46]. Therefore, it is possible that female offspring exposed to maternal obesity have lower systemic TNF-α, IL-6, and IL-1β related to lower activity levels compared to control offspring or male offspring exposed to maternal obesity.

We report longitudinal decreased IL-6 gene expression in the hypothalamus of male offspring exposed to maternal obesity compared to controls. Although increased hypothalamic IL-6 has been associated with an obesogenic phenotype, there is also a growing body of literature suggesting that hypothalamic IL-6 may be involved in mechanisms that are protective against obesity [32,47]. For example, hypothalamic IL-6 expression is lower in mice that are genetically prone to obesity, and IL-6 knock-out mice have increased body weight and decreased appetite stimulating neuropeptide (*Pomc*) expression in the hypothalamus [32]. Cytokines exert a neurotrophic effect in the developing brain, and specifically exogenous IL-6 has been shown to induce hypothalamic neurogenesis in mice [32]. Cytokine-induced apoptosis has also been described as a mechanism to eliminate overabundant neurons [14,32]. Therefore, it is plausible that the downregulation of hypothalamic IL-6 gene expression observed in offspring exposed to maternal obesity alters patterns of hypothalamic neurogenesis and apoptosis.

TNF-α, IL-6, and IL-1β induce anorexia through hypothalamic appetite signaling [43]. These cytokines induce appetite and weight loss when administered peripherally or directly into the brain [48,49,50]. Mechanistically, previous work has shown that decreased appetite is caused by decreases in corticotropin-releasing factor (CRF) and the stimulation of orexigenic neuropeptide Y (NPY) [50]. We report decreased hypothalamic TNF-α and IL-6 gene expression in neonatal offspring exposed to maternal obesity. Lower levels, which would theoretically increase appetite, may be an adaptation to the exposure to excess macronutrients from maternal obesity. Future studies to test this hypothesis would be important to investigate the relationship between early hypothalamic cytokine expression and energy homeostasis later in life.

We found lower TNF-α gene expression in the neonatal but not adult hypothalamus of offspring exposed to maternal obesity. We have previously shown, through pathway analysis, that the TNF gene was dysregulated in the hypothalamus of adult offspring exposed to maternal obesity and that TNF expression was moderated by sex [6]. As TNF is involved in the differentiation of neuroglia and the growth of neurites, one hypothesis is that decreased TNF levels may impair hypothalamic growth and differentiation, causing aberrant energy homeostasis leading to development of energy excess in adulthood [37,50].

Previous work demonstrates hypothalamic inflammatory changes in adults fed an obesogenic diet [14,17,18,19,20], as well as in offspring following exposure to maternal diet-induced obesity during hypothalamic development [6,21,22,23,38,51]. Furthermore, hypothalamic inflammation is decreased in diet-induced obesity-resistant WSB/EiJ mice compared to obesity-prone C57Bl/6J mice, underscoring the association between hypothalamic inflammation and the development of diet-induced obesity [52]. In contrast, the increased hypothalamic gene expression of TNF-α, IL-6, and IL-1β was not observed in our adult OB offspring despite increased body weight and adiposity. Since leptin itself has been shown to impact inflammation in mouse glial cells [22,53], we speculate that normal leptin levels in our study prohibited findings of hypothalamic inflammation.

Consistent with known sex differences in immune and neuroimmune activity, we report significant differences in peripheral cytokine levels in only female offspring exposed to maternal obesity and significant differences in hypothalamic cytokine expression in only male offspring exposed to maternal obesity. Prior studies have indicated that females have a greater peripheral immune response compared to males, while males have a greater neuroimmune response compared to females [54,55]. Sex chromosome genes and sex hormones including estrogen, progesterone, and androgens contribute to the differential regulation of immune response between sexes [54,55]. Of note, a study by Morselli et al. demonstrated increased hypothalamic inflammation in male wild-type mice compared to females in the setting of a high-fat diet, as well as increased inflammatory cytokines in estrogen receptor knock-out female mice [56]. Ongoing investigation of the intersection between biological sex and how environmental factors such as exposure to maternal obesity alter the development and function of the immune system is critical to developing therapies to prevent obesity.

There are several important limitations to our study that warrant further investigation. First, a limited number of cytokines were studied. While these target cytokines were chosen based on their strong, established relationships with obesity, a large breadth of study may offer novel insights or a more complete analysis of the relationship between developmental obesity and immune function. Second, this study is limited by solely evaluating hypothalamic mRNA expression. To better understand cytokine expression, investigating protein and cytokine secretion in hypothalamus is necessary. Third, the significance of our results may have been limited by a relatively small sample size. Additionally, further studies investigating immune cell differentiation and cell tracking would improve our understanding of how developmental stress, such as maternal obesity, affects immune function in the periphery and hypothalamus. Last, immune response is a complex interaction between tissue type, the dose of the stressor, the type of immune challenge, the timing of the stressor, the chronicity of the stressor, the hormonal state, the environment, and the biological sex. While this study considers many of these variables, further investigations that include varying iterations of these complex interactions are necessary prior to development of therapies that may prevent obesity with a developmental origin.

## 5. Conclusions

In conclusion, results from this study provide further support to research aimed at understanding how developmental inflammatory stress alters long-term health. A better understanding of how developmental stress intersects with biological sex, neurodevelopment, endocrine hormone activity, and immune function may lead to novel prevention and treatment strategies to mitigate the obesity epidemic.

## Figures and Tables

**Figure 1 biology-13-00399-f001:**
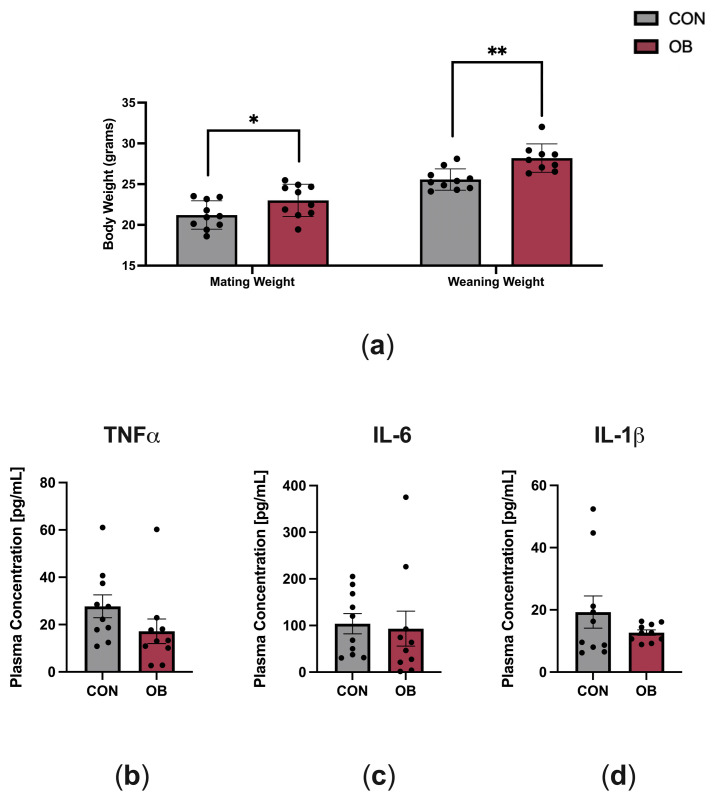
Characteristics of dams fed CON or OB diets. (**a**) Body weights at mating and weaning were significantly higher in OB versus CON dams. (**b**–**d**) Plasma cytokines measured on postnatal day 21 (TNF-α, IL-6, IL-1β) did not significantly differ between CON and OB dams. Results expressed as mean ± SEM; significance determined by unpaired *t*-test, * *p* < 0.05, ** *p* < 0.01. Gray bars: CON. Red bars: OB. N = 9–10/group.

**Figure 2 biology-13-00399-f002:**
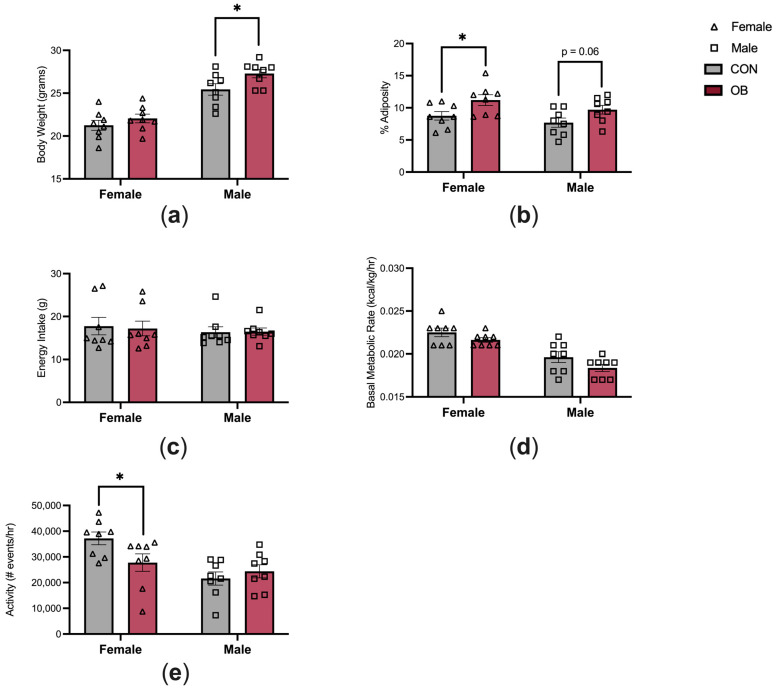
Maternal obesity alters offspring energy balance. (**a**) OB males are heavier than CON males, and (**b**) OB females have increased adiposity compared to CON females. (**c**,**d**) Energy intake and BMR are similar between groups. (**e**) OB female offspring have decreased activity compared to CON females. Results expressed as mean ± SEM; significance determined by unpaired *t*-test with males and females analyzed separately. * *p* < 0.05. Open triangles: female offspring. Open squares: male offspring. Gray bars: CON offspring. Red bars: OB offspring. N = 8/group/sex.

**Figure 3 biology-13-00399-f003:**
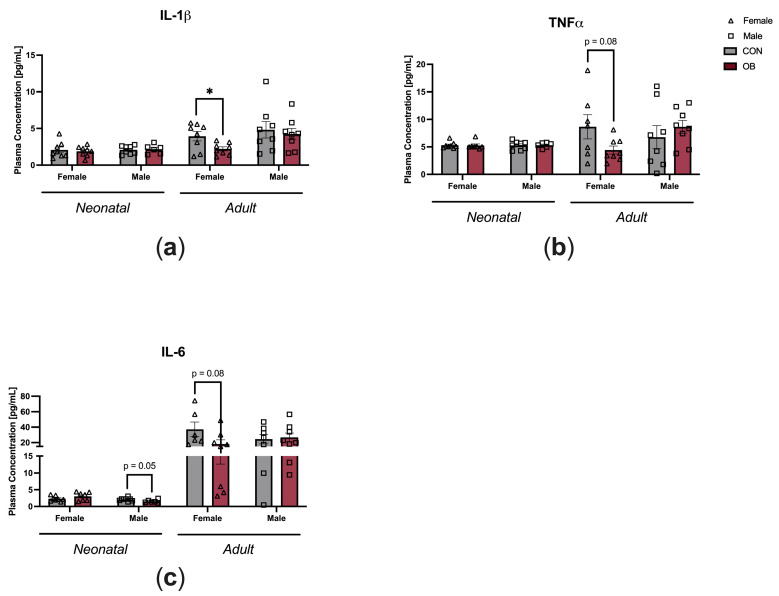
Maternal obesity is associated with decreased plasma cytokine levels in adult female offspring and decreased plasma IL-6 in neonatal male offspring. Plasma concentrations of (**a**) IL-1β, (**b**) TNF-α, and (**c**) IL-6 measured in neonatal and adult offspring of CON and OB dams. Results expressed as mean ± SEM; significance determined by unpaired *t*-test, * *p* < 0.05. Open triangles: female offspring. Open squares: male offspring. Gray bars: CON offspring. Red bars: OB offspring. N = 6–8/group.

**Figure 4 biology-13-00399-f004:**
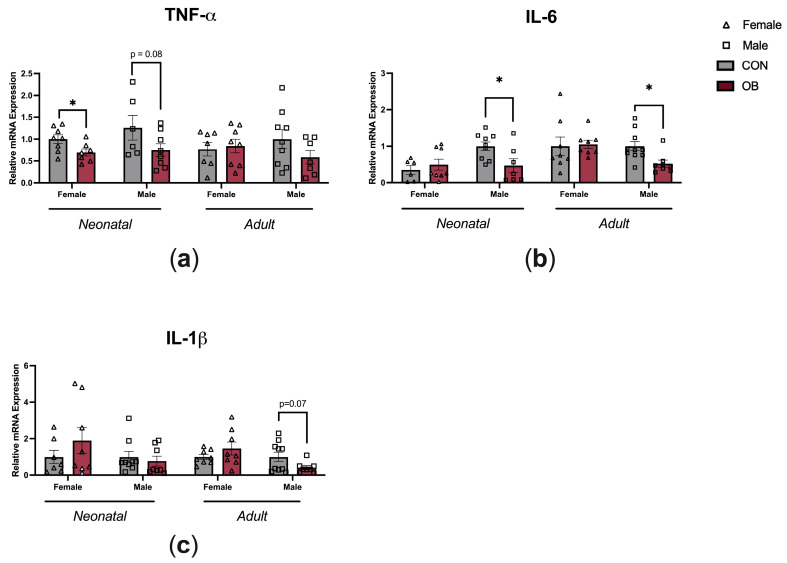
Maternal OB diet downregulates hypothalamic cytokine expression in neonatal and adult offspring in a sex-specific manner. Hypothalamic gene expression of (**a**) TNF-α, (**b**) IL-6, and (**c**) IL-1β measured in neonatal and adult offspring of CON and OB dams. Results expressed as mean ± SEM; significance determined by unpaired *t*-test, * *p* < 0.05. Open triangles: female offspring. Open squares: male offspring. Gray bars: CON offspring. Red bars: OB offspring. N = 6–10/group.

**Table 1 biology-13-00399-t001:** Adult offspring metabolic profile.

	CON (N = 16)	OB (N = 16)	*p*-Value
Insulin (ng/mL)	2.76 ± 0.3	2.24 ± 0.3	0.26
Leptin (ng/mL)	4.88 ± 1.3	6.67 ± 1.7	0.25
Cholesterol (mg/dL)	102.5 ± 6.5	102.4 ± 8.3	0.99
NEFA (mEq/L)	0.85 ± 0.07	0.81 ± 0.04	0.66
Triglycerides (mg/dL)	54.3 ± 6.6	50.9 ± 6.1	0.51
Phospholipids (mg/dL)	189.5 ± 11.3	196.9 ± 12.7	0.67

Data presented as mean ± SEM; significance determined by unpaired *t*-test.

## Data Availability

The data presented in this study are available on request from the corresponding author. The data are not publicly available due to privacy restrictions and ongoing projects.

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
