# Peer review of "Inflammation as a Sex-Specific Mediator in the Relationship between Maternal and Offspring Obesity in C57Bl/6J Mice"

_biology, 2024, doi:10.3390/biology13060399_

Round 1
Reviewer 1 Report (Previous Reviewer 2)
Comments and Suggestions for Authors
The article has partially made the necessary corrections and can be published in its current form.
Author Response
Thank you for your re-review of our manuscript. We appreciate your initial feedback and are pleased that you found our revised manuscript acceptable for publication.
Reviewer 2 Report (Previous Reviewer 3)
Comments and Suggestions for Authors
The authors' revisions have satisfactorily addressed my comments.
Author Response
Thank you for your re-review of our manuscript. We appreciate your initial feedback and are pleased that you found our revised manuscript acceptable for publication.
Reviewer 3 Report (New Reviewer)
Comments and Suggestions for Authors
The manuscript by Buckley et al., was a pleasure to read. The authors aim to outline hypothalamic alterations in neonates & adults in a model of maternal obesity. The authors use a milder model than most of the published literature available, but outline such correctly in the discussion. The results indicate that Dams did not experience the previously reported chronic low-grade inflammation measured as plasma levels of TNF etc. Throughout the experiments performed, males differ from females consistently. Energy expenditure is merely altered in females while males show an increase in weight. While female neonates do not show alterations in plasma inflammatory markers presented, males show a decrease in Il6. On the other hand, only females show significant reductions in Tnfa plasma levels as adults, while males remain unaltered. The effect on the hypothalamus seemed more severe compared to plasma analysis. Male and female neonates share a reduction in Tnfa in the neonate cohort. Conversely to plasma analysis, only males show IL-6 decreases as adults. Il1 beta was decreased in male neonates and adult females.
The mild model utilized in this study makes this manuscript particularly interesting and highlights a different state of MO and the effect on hypothalamic inflammation. This manuscript significantly contributes to the early stage of research in maternal obesity and how it alters brain, specifically hypothalamic tissue, inflammation.
A few comments should be addressed prior to publication:
Major comment:
The authors measure protein levels of inflammatory cytokines in the plasma of animals but only show genetic analysis of inflammatory gene expression for the hypothalamus. Given that the manuscript aims to uncover inflammatory changes specifically in the hypothalamus, these results should be validated by ELISA of tissue collected, Western or through ELISA on cerebral spinal fluid. These results will strongly increase the significance of the work presented and strengthen the claims stated in this manuscript.
Minor comment:
It would be helpful to introduce the role of TNF, IL1 beta and IL6 also in the introductions given their importance in the study and why they were chosen. Should be concise and brief.
Author Response
Major comment:
The authors measure protein levels of inflammatory cytokines in the plasma of animals but only show genetic analysis of inflammatory gene expression for the hypothalamus. Given that the manuscript aims to uncover inflammatory changes specifically in the hypothalamus, these results should be validated by ELISA of tissue collected, Western or through ELISA on cerebral spinal fluid. These results will strongly increase the significance of the work presented and strengthen the claims stated in this manuscript.
Author Response:
Thank you for this important feedback. We agree that the significance of our findings would be increased if we were able to present both gene and protein expression of cytokines in the hypothalamus. We hope this paper provides an opportunity to inspire future work within our lab and others. At this time, we do not have the hypothalamic tissue available to do these experiments. From a feasibility standpoint the animal model would require another 9-months to allow for those experiments. However, we do acknowledge the reviewer’s important point and included this statement within the limitations of the paper (lines 310-312). While we agree that the lack of protein quantification is a limitation of our study, we maintain that our findings have merit on their own and we believe that the foundational work described here is worthy of publication. As the reviewer highlighted in their comments, our manuscript is unique in that it is the first characterization of inflammation in our specific mouse model, which produces a comparatively milder phenotype of maternal obesity than what is commonly reported in the literature. While further investigation is necessary to fully characterize mechanisms of inflammation in offspring exposed to maternal diet induced obesity, this paper is a necessary first report describing both longitudinal and sex-specific findings that expand upon previous knowledge of this established mouse model.
Minor comment:
It would be helpful to introduce the role of TNF, IL1 beta and IL6 also in the introductions given their importance in the study and why they were chosen. Should be concise and brief.
Author Response:
We appreciate this comment. We agree that it would be helpful to introduce the role of TNF, IL-1 beta and IL6 sooner in the manuscript. We have relocated the following sentence that describes why these specific cytokines were chosen from the Methods section (L114) to the Introduction section (L75): “TNF-α, IL-1β, and IL-6 were chosen as inflammatory targets as they are most commonly associated with and reported in obesity literature [15,24,29-36].”
Round 2
Reviewer 3 Report (New Reviewer)
Comments and Suggestions for Authors
The authors have addressed the major and minor concerns sufficiently by editing the manuscript's limitations and introduction. No further comments.
Author Response
Thank you very much; we appreciate your feedback.
This manuscript is a resubmission of an earlier submission. The following is a list of the peer review reports and author responses from that submission.
Round 1
Reviewer 1 Report
Comments and Suggestions for Authors
In this manuscript, Buckley et al. have evaluated whether high-fat feeding during pregnancy and lactation distinct affect systemic and hypothalamic pro-inflammatory cytokine expression in male and female littermates both in neonatal and adult periods. The authors observed some differences in the hypothalamic mRNA expression and plasma expression of IL-6, IL-1-beta, and TNF-alpha, in both sexes and periods of life, in comparison with offspring of control dams (i.e. fed exclusively on chow diet).
The aim of the study is interesting, but the manuscript is very simple and does not significantly contribute to the advancement of knowledge in this field of study.
Regarding the methods, the author just employed qPCR and ELISA assays for assessing cytokines expression in the hypothalamus and plasma, respectively. There are also some metabolic measurements, such as locomotor activity, energy intake and expenditure, body mass, and adiposity, and the analysis of hormones and other biochemical markers in the plasma, but even altogether, these analyses do not increase the relevance of the study.
Briefly, this is just a descriptive paper that shows changes in the expression of cytokines in the CNS and systemically, but there are plenty of analyses missing for the conclusion presented: "Based on our findings, we conclude that maternal obesity is associated with excess energy balance in offspring and may be, at least partially, mediated by aberrant immune functioning in a sex specific manner."
Based on the several flaws present above, I do not recommend this article for publishing in Biology (MDPI) in this present form.
Reviewer 2 Report
Comments and Suggestions for Authors
The study should be evaluated from its histopathological perspective. TNF-α, IL-1β, and IL-6 immunolocalizations in tissues should be determined. Obesity status should be visualized histologically. Introduction and material method should be arranged accordingly. An animal ethics committee decision should be added to the study. The discussion should be shortened. Too many references are used, the number of references should be reduced. Especially references from the last 5 years should be included. The study should be re-evaluated after the necessary changes have been made.
Reviewer 3 Report
Comments and Suggestions for Authors
The manuscript by Buckley et al. entitled:
“Inflammation as a Sex Specific Mediator in the Relationship between Maternal and Offspring Obesity in C57Bl/6J Mice”
is an original study addressing the effects of maternal obesity on metabolic and inflammatory markers of the offspring.
It is a well-driven experimental study addressing obesity, a current health issue of social concern. Results provide relevant information that could eventually complement other studies in order to neuroprotect offspring from developmental obesity.
Thus, I recommend it for publication.
Yet, following comments should be considered:
L161 “Plasma cytokines measured on postnatal day 21 (TNF-α, IL-6, IL-1β) did not differ” You may want to include significantly before differ, since there seems to be a tendency.
L220: On “Discussion” you may want to compare results and include references from other comparative studies such as: Terrien et al, 2019 (doi: 10.1038/s41598-019-56051-4), and Morselli et al, 2016 (DOI: 10.1038/ijo.2015.114 ).
L303: “There are several important limitations to our study that warrant further investiga-303 tion”. You may want to include the n as a limitation, since probably some parameters as TNF (referred in L161), could be significant if higher n would have been studied.
L335: “to end the obesity epidemic” you may want to substitute “end” by “mitigate”.
